# Federated Hypergradient Descent

## Abstract

In this work, we explore combining automatic hyperparameter tuning and optimization for federated learning (FL) in an online, one-shot procedure. We apply a principled approach on a method for adaptive client learning rate, number of local steps, and batch size. In our federated learning applications, our primary motivations are minimizing communication budget as well as local computational resources in the training pipeline. Conventionally, hyperparameter tuning methods involve at least some degree of trial-and-error, which is known to be sample inefficient. In order to address our motivations, we propose FATHOM (Federated AuTomatic Hyperparameter OptiMization) as a one-shot online procedure. We investigate the challenges and solutions of deriving analytical gradients with respect to the hyperparameters of interest. Our approach is inspired by the fact that we have full knowledge of all components involved in our training process, and this fact can be exploited in our algorithm impactfully. We show that FATHOM is more communication efficient than Federated Averaging (FedAvg) with optimized, static valued hyperparameters, and is also more computationally efficient overall. As a communication efficient, one-shot online procedure, FATHOM solves the bottleneck of costly communication and limited local computation, by eliminating a potentially wasteful tuning process, and by optimizing the hyperparamters adaptively throughout the training procedure without trial-and-error. We show our numerical results through extensive empirical experiments with the Federated EMNIST-62 (FEMNIST) and Federated Stack Overflow (FSO) datasets, using FedJAX as our baseline framework.

## 1 Introduction

Federated learning (FL) for on-device applications has its obvious social implications, due to its inherent privacy-protection feature. It opens up a broad range of opportunities to allow a massive number of devices to collaborate in developing a shared model by retaining private data on the devices. The ubiquity of machine learning (ML) on consumer data, coupled with the growth of privacy concerns, has pushed researchers and developers to look for new ways to protect and benefit end-users. In order for FL to deliver its promise in deployed applications, there are still many open challenges remained to be solved. We are especially interested in the overall communication efficiency of the FL pipeline for it to be realistically deployed in a unique communication environment over expensive links. To begin, consider a typical step in a machine learning (ML) pipeline: hyperparameter tuning. Whether it is in a centralized, distributed or federated setting, it is an essential step to achieve an optimal operation for the training process. At the heart of an ML training process is the optimization algorithm. In particular, we are interested in using Federated Averaging (FedAvg) as our baseline

federated optimization algorithm for our work. This is because, despite all the recent innovations in FL since its introduction in 2016 by McMahan et al. [2016], FedAvg remains the de facto standard in federated optimization for both research and practice, due to its simplicity and empirical effectiveness. In order for FedAvg to operate effectively, it requires properly tuned hyperparameter values.

Our work focuses specifically on hyperparameter optimization (HPO) of: 1) client learning rate, 2) number of local steps, as well as 3) batch size, for FedAvg. We propose FATHOM (Federated AuTomatic Hyperparameter OptiMization), which is an online algorithm that operates as a one-shot procedure. In the rest of this paper, we will go through a few notable recent state-of-the-art works on this topic, and make justifications for our new approach. Then we will derive a few key steps for our algorithm, followed by a theoretical convergence bound for adaptive learning rate and number of local steps in the non-convex regime. Lastly, we present numerical results on our empirical experiments with neural networks on the FEMNIST and FSO datasets.

Our contributions are as follows:

- We derive gradients with respect to client learning rate and number of local steps for FedAvg, for an online optimization procedure. We propose FATHOM, a practical one-shot procedure for joint-optimization of hyperparameters and model parameters, for FedAvg.

- We derive a new convergence upper-bound with a relaxed condition (see Section 4 and remark 2), to highlight the benefits from the extra degree-of-freedom that FATHOM delivers for performance gains.

- We present empirical results that show state-of-the-art performance. To our knowledge, we are the first to show gain from an online HPO procedure over a well-tuned equivalent procedure with fixed hyperparameter values.

## 2 Related Work and Justifications for FATHOM

We explore the question whether the FATHOM approach is justified over the more recent, state-of-the-art methods that are designed for the same goal: a single-shot online hyperparamter optimization procedure for FL. Zhou et al. [2022] proposed Federated Loss SuRface Aggregation (FLoRA), a general single-shot HPO for FL, which works by treating HPO as a black-box problem and by performing loss surface aggregation for training the global model. Khodak et al. [2021] draws inspiration from weight-sharing in Neural Architectural Search (Pham et al. [2018], Cai et al. [2019]), and proposed FedEx, which is an online hyperparameter tuning algorithm that uses exponentiated gradients to update hyperparameters. On the other hand, Mostafa [2019]'s RMAH and Guo et al. [2022]'s Auto-FedRL both use REINFORCE (Williams [1992]) in their agents to update hyperparameters in an online manner, by using relative loss as their trial rewards. One basic assumption among these methods, is that at least some of the gradients with respect to the hyperparameters are unavailable directly. Generalized techniques are used to update these quantities, involving Monte-Carlo sampling and evaluation with held-out data. One key benefit with techniques such as these is their generalizability for a wide range of different hyperparameters. On the other hand, we identify a few areas with these methods that we would like to improve on. One, information about the internals of the procedure can and should be exploited. Two, communication overhead becomes a concern, since sufficient Monte-Carlo sampling is required for some of these techniques to converge, an example being the re-parametrization trick (Kingma and Welling [2013]) which is used for FedEx, RMAH and Auto-FedRL. From initial observations of their empirical results, while these methods are successful in hyperparameter tuning and reaching target model accuracy as shown in these works, these goals are achieved in unspecified numbers of total communication rounds from works based on RL approaches such as Mostafa [2019] and Guo et al. [2022].

The above observations justify exploring our problem differently from previous approaches. Our method exploits full knowledge of the training process, and it does not require sufficient trials at potential expense of communication budget. Inspired by the hypergradient descent techniques developed by Baydin et al. [2017] and Amid et al. [2022] for centralized optimization learning rate, we develop FATHOM by directing deriving analytical gradients with respect to the hyperparameters of interest. The result is a sample efficient method which offers both improvements in communication efficiency and reduced local computation in a single-shot online optimization procedure. Meanwhile, FATHOM is not as flexibly applicable in optimizing a wide range of hyperparameters, since each

89  gradient needs to be derived separately to take advantage of our full knowledge of the training process.
90  We believe this approach is a performance advantage, at the expense of its flexibility.

91  There are other notable relevant works. Charles and Konečný [2020] and Li et al. [2019] proved that
92  reducing the client learning rate during training is necessary to reach the true objective. Yet, a line of
93  interesting works, such as Dai et al. [2020] and Holly et al. [2021]) applies Bayesian Optimization
94  (BO) on federated hyperparamter tuning, by treating it as a closed-box optimization problem. Dai
95  et al. [2021] further updates their use of BO in FL by incorporating differential privacy. However,
96  these BO-based works do not consider adaptive hyperparameters. Yet, another work (Wang and Joshi
97  [2018]) shares similarity to our approach of optimally adapting the number of local steps, with their
98  adaptive communication strategy, AdaComm, in the distributed setting. However, their main interest
99  is reducing wall-clock time. Lastly, around the same time of this writing, Wang et al. [2022] publishes
100 their benchmark suite for FL HPO, called FedHPO-B, which would be valuable to our future work.

## 3   Methodology

102 In this section we formalize the problem of hyperparameter optimization (HPO) for FL. We first
103 review FedAvg, a de facto standard of federated optimization methods for research baseline and
104 practice. Then, we present our method for online-tuning of its hyperparameters, specifically client
105 learning rate, number of local steps, and batch size. We call our method FATHOM (Federated
106 AuTomatic Hyperparameter OptiMization).

### 3.1   Problem Definition

108 In this paper, we consider the empirical risk minimization (ERM) across all the client data, as an
109 unconstrained optimization problem:

$$f^* := \min_{x \in \mathbb{R}^d} \left[ f(x) := \frac{1}{m} \sum_{i=1}^{m} f_i(x) \right] \tag{1}$$

110 where $f_i : \mathbb{R}^d \to \mathbb{R}$ is the loss function for data stored in local client index $i$ with $d$ being the
111 dimension of the parameters $x$, $m$ is number of clients, and $f^* = f(x_*)$ where $x_*$ is a stationary
112 solution to the ERM problem in eq(1).

113 To facilitate some of the discussions that follow, it helps to define assumptions here as we do
114 throughout the rest of this paper:

115 **Assumption 1.** *(Unbiased Local Gradient Estimator) Let $g_i(x)$ be the unbiased, local gradient*
116 *estimator of $\nabla f_i(x)$, i.e., $\mathbb{E}[g_i(x)] = \nabla f_i(x)$, $\forall x$, and $i \in [m]$.*

### 3.2   Federated Optimization and Tuning of Hyperparameters

118 **Federated Averaging (FedAvg)**   We describe the operations of FedAvg from McMahan et al.
119 [2016], as follows. At any round $t$, each of the $m$ clients takes a total of $K_i$ local SGD steps, where
120 $K_i = \lfloor E\nu_i/B \rfloor$, and where $\nu_i$ is the number of data samples from client index $i$, $B$ is batch size,
121 with epoch number $E = 1$ being a common baseline. In this version of FedAvg, heterogeneous data
122 size is accommodated across clients, and the number of local steps can be manipulated via $E$ and
123 $B$ as hyperparameters. Each local SGD step updates the local model parameters of each client $i$ as
124 follows: $x^i_{t,k+1} = x^i_{t,k} - \eta_L g_i(x^i_{t,k})$, where $\eta_L$ is the local learning rate and $k \in [K]$ is the local step
125 index. To conclude each round, these clients return the local parameters $x^i_{t,K_i}$ to the server where
126 it updates its global model, with $x_{t+1} = \sum_i \nu_i x^i_{t,K}/\nu$ where $\nu = \sum_i \nu_i$. To facilitate some of the
127 discussions that follow, we define the following quantities:

$$\overline{\Delta}_t \triangleq x_{t+1} - x_t = \sum_{i=1}^{m} \frac{\nu_i}{\nu} \Delta^i_t \quad \text{where} \quad \Delta^i_t \triangleq - \sum_{k=0}^{K_i-1} \eta_{L,t} g_i(x^{i,k}_t) \tag{2}$$

128 **Offline Hyperparameter Tuning**   Offline tuning is best to be summarized as follows. We first
129 define $U = \{u \in \mathbb{R} \mid u \geq 0\}$ with $\eta_L \in U$, and $V = \{v \in \mathcal{I} \mid v \geq 1\}$ with $K \in V$. We also define
130 $C = U \times V$, and $c = (\eta_L, K)$, where $c \in C$. Offline tuning would have the following objective:

$\min_{c \in C} f_{\text{valid}}(x, c)$ s.t. $x = \operatorname{argmin}_{z \in \mathbb{R}^d} f_{\text{train}}(z, c)$ . With abuse of notation, we use $f_{\text{valid}}$ for the objective function calculated from a validation dataset which is usually held-out before the procedure, and $f_{\text{train}}$ for the objective from training data which usually is just local client data. A few notable offline tuning methods are as follows. Global grid-search from Holly et al. [2021] is an example of offline tuning that iterates over the entire search grid defined as $C$, completing an optimization process for each grid point and evaluating the result with a held-out validation set. Global Bayesian Optimization from Holly et al. [2021] is another similar example of offline tuning that follows the same template and objective. Instead of brute-force grid-search, $c$ is sampled from a distribution $\mathcal{D}_C$ over $C$, i.e. $c \sim \mathcal{D}_C$, that updates after every iteration.

**Online Hyperparameter Optimization** We are interested in an online procedure that combines hyperparameter optimization and model parameter optimization, with the following objective:

$$\min_{\substack{x \in \mathbb{R}^d \\ c \in C}} f_{\text{train}}(x, c) \tag{3}$$

This formulation is the objective of our method, FATHOM, which we will discuss shortly in detail. It has the advantage of joint optimization in a one-shot procedure. Furthermore, it does not assume the availability of a validation dataset.

### 3.3 Our Method: FATHOM

In this section we will introduce our method, FATHOM (Federated AuTomatic Hyperparameter OptiMization). Recall from our joint objective, eq(3), that both the model parameters, $x$, and hyperparameters of the optimization algorithm, $c$, are optimized jointly to minimize our objective function. An alternative view is to treat $c$ as part of the parameters being optimized in a classic formulation, i.e. $min_y f(y)$ with $y = (x, c)$. As previously mentioned, our method is inspired by hypergradient descent from Baydin et al. [2017] and by exponentiated gradient from Amid et al. [2022], both proposed for centralized learning rate optimization. We will present how FATHOM exploits our knowledge of analytical gradients to update client learning rate, number of local steps, as well as batch size, for an online, one-shot optimization procedure.

**Assumption 2.** *(Convexity w.r.t. $\eta_L$ and $K$) We assume $\mathbb{E}_t(f(x_t))$ is convex w.r.t. $\eta_L$ and $K$, even though we assume non-convexity w.r.t. $x_t$). Specifically, convexity w.r.t. $K$ follows the definition in Murota [1998], to accommodate the integer space where $K$ is defined.*

**Remark 1.** *Assumption 2 is necessary to guarantee the existence of subgradients derived in Theorems 1 and 2, and it will be assumed for this work. In problems dealing with deep neural networks, it is reasonable to not assume convexity w.r.t. hyperparameters. However, from our empirical results, we claim that the proposed algorithm is still able to operate as desired under this condition.*

#### 3.3.1 Hypergradient for Client Learning Rate

In this section, we derive the hypergradient for client learning rate in a similar fashion as Baydin et al. [2017], with the difference being that they are mainly concerned with the centralized optimization problem, and that we are concerned with the distributed setting where clients take local steps. We derive the following hypergradient of the objective function as defined in eq(1), taken with respect to the learning rate $\eta_{L,t-1}$ such that it can be updated to obtain $\eta_{L,t}$:

$$H_t = \frac{\partial f(x_t)}{\partial \eta_{L,t-1}} = \frac{\partial f(x_t)}{\partial x_t} \cdot \frac{\partial (x_{t-1} + \overline{\Delta}_{t-1})}{\partial \eta_{L,t-1}} = \nabla f(x_t) \cdot \frac{\partial \overline{\Delta}_{t-1}}{\partial \eta_{L,t-1}} \tag{4}$$

where $\overline{\Delta}_t$ is the update step for the global parameters $x_t$ as defined in eq(2), leading to $\frac{\partial \overline{\Delta}_t}{\partial \eta_{L,t}} = \frac{\overline{\Delta}_t}{\eta_{L,t}} = -\sum_{i=1}^{m} \frac{\nu_i}{\nu} \sum_{k=0}^{K-1} g_i(x_t^{i,k})$. We also make the approximation $x_{t+1} - x_t = \overline{\Delta}_t \approx -\eta_{L,t} \nabla f(x_t)$. We can then write the normalized update, $\overline{H}_t$, similar to Amid et al. [2022], as follows:

$$\overline{H}_t = \frac{\nabla f(x_t)}{\|\nabla f(x_t)\|} \cdot \left( \frac{\partial \overline{\Delta}_{t-1}}{\partial \eta_{L,t-1}} \Big/ \Big\| \frac{\partial \overline{\Delta}_{t-1}}{\partial \eta_{L,t-1}} \Big\| \right) \approx -\frac{\overline{\Delta}_t}{\|\overline{\Delta}_t\|} \cdot \frac{\overline{\Delta}_{t-1}}{\|\overline{\Delta}_{t-1}\|} \tag{5}$$

The resulting hypergradient is a scalar, as expected, and can be used efficiently as part of the update rule for $\eta_L$, which we will see in Section 3.3.4. The implementation is communication efficient, since in each round, each client needs one extra scalar to send back to the server, and likewise the server needs to broadcast one extra scalar back to the clients. It is also computationally efficient since it avoids calculating the full local gradient $\nabla f(x_t)$.

### 3.3.2 Hypergradient for Number of Local Steps

Since the number of local steps is an integer, i.e. $K = \{k \in \mathbb{I} \mid k \geq 1\}$, this means $f(x_t)$ does not exist for non-integer values of $K$. We formulate a subgradient as a surrogate of the hypergradient $\partial f(x_t)/\partial K$, as follows. We will call this a hyper-subgradient.

**Theorem 1.** *When a piecewise function $L_t$ is defined for every value of $K_0 \in [K]$ on $l$, such that $0.0 \leq l < 1.0$, we claim, under Assumption 2, that the following is a subgradient of $f(x_t)$ at $K_t = K_0$:*

$$\frac{\partial L_t}{\partial l} = \nabla f(x_t) \cdot \Big( -\eta_{L,t} \sum_{i=1}^{m} g_i(x_{t-1}^{i,K_t-1}) \frac{\nu_i}{\nu} \Big) \tag{6}$$

*where $l$ represents the marginal fraction of local steps beyond $K_0$. We leave the proof (with an illustration in Figure 2) in the Appendix section beginning in eq(20).*

The result from Theorem 1 is not sufficiently communication-efficient for implementing an update rule for $K$. This is because it would require the quantity $g_i(x_{t-1}^{i,K_t-1})$ to be communicated from each client $i$ to the server. To save communication, let us reuse what the server has in memory: $\overline{\Delta}_t = \Big( -\eta_L \sum_{i=1}^{m} \frac{\nu_i}{\nu} \sum_{k=0}^{K_t-1} g_i(x_t^{i,k}) \Big)$. If we let:

$$S_t = \nabla f(x_t) \cdot \Big( -\eta_{L,t} \sum_{i=1}^{m} \frac{\nu_i}{\nu} \sum_{k=0}^{K_t-1} g_i(x_{t-1}^{i,k}) \Big) l \tag{7}$$

$$N_t = \frac{\partial S_t}{\partial l} = \nabla f(x_t) \cdot \Big( -\eta_{L,t} \sum_{i=1}^{m} \frac{\nu_i}{\nu} \sum_{k=0}^{K_t-1} g_i(x_{t-1}^{i,k}) \Big) = \nabla f(x_t) \cdot \overline{\Delta}_{t-1} \tag{8}$$

$$\overline{N}_t = \frac{\nabla f(x_t)}{\|\nabla f(x_t)\|} \cdot \frac{\overline{\Delta}_{t-1}}{\|\overline{\Delta}_{t-1}\|} \approx -\frac{\overline{\Delta}_t}{\|\overline{\Delta}_t\|} \cdot \frac{\overline{\Delta}_{t-1}}{\|\overline{\Delta}_{t-1}\|} \tag{9}$$

where eq(9) is the normalized update as in Amid et al. [2022]. We claim that eq(8) is a positively-biased version of eq(6), which has its practical importance due to the fact that the last term in eq(6) from Theorem 1 results in zero-mean, noisy gradients, when the local functions are nearing their local solutions, when in fact, this is the area where more local work is not needed. Thus, a positive bias is desirable to drive the number of local steps down. This result is also useful from a communication efficiency perspective in its implementation, because the server has all the components to calculate this quantity, and would not require additional communication.

### 3.3.3 Regularization for Number of Local Steps

One of the goals for FATHOM is savings in local computation. To avoid excessive number of local steps, we further develop a regularization term for local computation against excessive $K$, which is a proxy for the hypergradient of the local client functions at the end of each round : $\partial f_i(x_t^{i,K})/\partial K$.

**Theorem 2.** *When a piecewise function $J_t$ is defined for every value of $K_0 \in [K]$ on $l$, such that $0.0 \leq l < 1.0$, we claim, under Assumption 2, that the following is a subgradient of $\sum_{i=1}^{m} f_i(x_t^{i,K_t})$ at $K_t = K_0$:*

$$\frac{\partial J_t}{\partial l} = -\eta_{L,t} \sum_{i=1}^{m} \frac{\nu_i}{\nu} \mathbb{E}\big[g_i(x_t^{i,K_0-1})\big] \cdot g_i(x_t^{i,K_t}) \approx -\eta_{L,t} \sum_{i=1}^{m} \frac{\nu_i}{\nu} \sum_{k=0}^{K_t-1} g_i(x_t^{i,k}) \cdot g_i(x_t^{i,K_t}) \tag{10}$$

*where $l$ represents the marginal fraction of local steps beyond $K_0$. We leave the proof in the Appendix section beginning in eq(24).*

In our algorithm, we use the normalized update based on the following biased proxy, since eq(10) tends to be noisy from $g_i(x_t^{i,K_t})$.

$$G_t = -\eta_{L,t} \sum_{i=1}^{m} \frac{\nu_i}{\nu} \min_{K \leq K_t} \Big( \sum_{k=0}^{K-1} g_i(x_t^{i,k}) \cdot g_i(x_t^{i,K}) \Big) \tag{11}$$

$$\overline{G}_t = -\eta_{L,t} \sum_{i=1}^{m} \frac{\nu_i}{\nu} \min_{K \leq K_t} \left( \frac{\sum_{k=0}^{K-1} g_i(x_t^{i,k})}{\big\| \sum_{k=0}^{K-1} g_i(x_t^{i,k}) \big\|} \cdot \frac{g_i(x_t^{i,K})}{\|g_i(x_t^{i,K})\|} \right) \tag{12}$$

where $\overline{G}_t$ is the normalized update. The proxy yields a bias towards smaller number of local steps, which is desirable for reducing local computation. We use this biased proxy against using a more typical regularization such as L2 for the number of local steps, based on initial empirical results for better performance..

### 3.3.4 Normalized Exponentiated Gradient Updates

For the update rules of the hyperparameters $\eta_L$ (client learning rate) and $K$ (number of client local steps), we use the normalized exponentiated gradient descent method (EGN) with no momentum, rather than a conventional linear update method such as the additive update of hypergradient descent proposed in Baydin et al. [2017]. It is reasonable to use exponentiated gradient (EG) methods for updates of hyperparameters that are strictly positive in value. EG methods also enjoy significantly faster convergence properties when only a small subset of the dimensions are relevant, according to Amid et al. [2022].

EG methods have been proposed in previous works for a variety of applications (Khodak et al. [2021], Amid et al. [2022], Li et al. [2020]), and analyzed in depth (Ghai et al. [2019]), where its convergence has been studied and validated (Li and Cevher [2018]). Recently, Amid et al. [2022] showed that EGN is the same as the multiplicative update for hypergradient descent proposed in Baydin et al. [2017], when the approximation $exp(\cdot) \approx 1 + \cdot$ is made. From our observations, we believe that momentum is not needed for the effectiveness of EGN in our application, as validated in our numerical results. We also opted-out of adding further complexity such as extra weights and activation functions to model the relationships between $\eta_{L,t}$ and $K_t$, because it would require more samples to optimize and because FATHOM is a one-shot procedure. Furthermore, due to the non-stationary nature of these values, we opt for a simpler scheme for faster performance.

Hence, for the update rule of client learning rate, $\eta_L$, we have:

$$\eta_{L,t+1} = \eta_{L,t} \exp\left(-\gamma_\eta \overline{H}_t\right) \tag{13}$$

where $\overline{H}_t$ is as defined in eq(5). For number of local steps, we observe that it is related to batch size in round $t$, $B_t$, as follows. To accommodate heterogeneity of local dataset sizes among clients, we have number of local data samples from client $i$ to be $\nu_i$. The number of local steps for client $i$ is $K_i = \lfloor \nu_i E_t / B_t \rfloor$, where $E_t$ is number of epochs, with $E_t = 1$ meaning the entire local dataset for each client to be processed once per round. We derive update rules for $E_t$ and $B_t$ globally to optimize the number of local steps, without having to make any changes to our theoretical analysis to accommodate the heterogeneity of local dataset sizes:

$$E_{t+1} = E_t \exp\left(-\gamma_E\left(\overline{N}_t + \overline{G}_t\right)\right) \tag{14}$$

and

$$B_{t+1} = B_t \exp\left(-\gamma_B\left(-\overline{G}_t\right)\right) \tag{15}$$

where $N_t$ and $G_t$ are defined in eq(9) and eq(12), respectively. These update rules accomplish the goal of updating the number of local steps via $E_t/B_t$ with $\frac{E_{t+1}}{B_{t+1}} = \frac{E_t}{B_t} \exp\left(-\gamma_E \overline{N}_t - \left(\gamma_E - \gamma_B\right)\overline{G}_t\right)$. Typically, with $\gamma_B \geq \gamma_E$, $(\gamma_B - \gamma_E)\overline{G}_t$ becomes a tunable regularization term as discussed at the end of Section 3.3.3.

### 3.3.5 Client Sampling

We present our method, FATHOM, as shown in Algorithm 1. One practical factor we have not considered in our discussions is partial client sampling. For our implementation to handle the stochastic nature of client sampling, the metric $\overline{\Delta}_{t-1}$ for calculating $\overline{H}_t$ in eq(5) and $\overline{N}_t$ in eq(9) is modified by a smoothing function for noise filtering, i.e. $\overline{\Delta}_{t,sm} = \alpha\overline{\Delta}_{t-1,sm} + (1-\alpha)\overline{\Delta}_t$, which is a single-pole infinite impulse response filter (Oppenheim and Schafer [2009]Oppenheimer et al. [2009]) with no bias compensation. We use the notation "sm" for smoothed, and after many experiments, we decide to use $\alpha = 0.5$ for all of our numerical results.

**Algorithm 1: FATHOM** : $g_i(x)$ is defined in Assumptions 1, and $m$ is the number of clients.

**Input:** Server initializes global model $x_{t=1}$, $T$ as the end communication round, and:

$$\overline{\Delta}_{t=0,sm} = 0 \; ; \; \alpha = 0.5 \; ; \; \gamma_\eta = 0.01 \; ; \; \gamma_E = 0.01 \; ; \; \gamma_B = 0.1$$

**Output:** $x_T$, as well as $\eta_{L,t}$, $E_t$ and $B_t$ for all $t \in [T]$

**for** $t = 1, \ldots, T$ **do**

    Sample client set $S_t$ out of $m$ clients.

    For each client $i \in S_t$, initialize: $x_t^{i,k=0} = x_t$ and $K_{t,i} = \lfloor \nu_i E_t / B_t \rfloor$ .

    Set $\Delta_i = 0$, and $\phi_i = +\infty$.

    **for** $k = 0, \ldots, K_{t,i} - 1$ **do**

        For each client $i$, compute in parallel an unbiased stochastic gradient $g_i(x_t^{i,k})$.

        For each client $i$, calculate $\phi_i = \min(\phi_i, g_i(x_t^{i,k}) \cdot \Delta_i)$ where $\Delta_i = x_t^{i,k} - x_t$

        For each client $i$, update in parallel its local solution: $x_t^{i,k+1} = x_t^{i,k} - \eta_{L,t} g_i(x_t^{i,k})$

    **end**

    Server calcualtes $\nu = \sum_{i \in S_t} \nu_i$, where $\nu_i$ is the size of client $i$ dataset.

    Server calculates $\overline{\Delta}_t = \sum_{i \in S_t} \Delta_i (\nu_i/\nu)$; see eq(2)

    Server updates global model $x_{t+1} = x_t - \overline{\Delta}_t$

    Server calculates $\overline{H}_t = \overline{N}_t = -\frac{\overline{\Delta}_t}{\|\overline{\Delta}_t\|} \cdot \frac{\overline{\Delta}_{t-1,sm}}{\|\overline{\Delta}_{t-1,sm}\|}$, modified from eq(5) and eq(9)

    Server calculates $\overline{G}_t$; see eq(12)

    Server updates client learning rate $\eta_{L,t+1}$, epochs, $E_{t+1}$, and batch size $B_{t+1}$ for the next
      round; see eq(13), eq(14), and eq(15).

    Server updates $\overline{\Delta}_{t,sm} = (1 - \alpha)\overline{\Delta}_t + \alpha\overline{\Delta}_{t-1,sm}$ for the next round

**end**

## 4 Theoretical Convergence

A standard approach to theoretical analysis of an online optimization method such as ours, is through analyzing the regret bound (Zinkevich [2003], Khodak et al. [2019], Kingma and Ba [2014], and Mokhtari et al. [2016]). Nonetheless, this approach does not tell us the impact on communication efficiency by the online updates introduced from FATHOM. Therefore, we take an alternative approach by extending the guarantees of FedAvg performance (Wang et al. [2021], Reddi et al. [2020], Gorbunov et al. [2020], Yang et al. [2021], Li et al. [2019], etc) to include both adaptive learning rate and adaptive number of local steps. We assume the special case in our analysis to have full client participation. We prove that adaptive learning rate and adaptive number of local steps does not impact asymptotic convergence, despite the given relaxed conditions.

### 4.1 Assumptions

**Assumption 3.** *(L-Lipschitz Continuous Gradient for Parameters $x_t$) There exists a constant $L > 0$, such that $\|\nabla f_i(x) - \nabla f_i(y)\| \leq L\|x - y\|$, $\forall x, y \in \mathbb{R}^d$, and $i \in [m]$, where $x$ and $y$ are the parameters in eq(1).*

**Assumption 4.** *(Bounded Local Variance) There exist a constant $\sigma_L > 0$, such that the variance of each local gradient estimator is bounded by $\mathbb{E}\|\nabla f_i(x) - g_i(x)\|^2 \leq \sigma_L^2$, $\forall x$, and $i \in [m]$.*

**Assumption 5.** *(Bounded Second Moment) There exists a constant $G > 0$, such that $\mathbb{E}_t\|\nabla f_i(x_t)\| \leq G$, $i \in [m]$, $\forall x_t$.*

### 4.2 Convergence Results

**Theorem 3.** *Under Assumptions 1-5 and with full client participation, when FATHOM as shown in Algorithm 1 is used to find a solution $x_*$ to the unconstrained problem defined in eq(1), the sequence of outputs $\{x_t\}$ satisfies the following upper-bound, where, with slight abuse of notation, $\mathcal{E} = \min_{t \in [T]} \mathbb{E}_t\|\nabla f(x_t)\|_2^2$:*

$$\mathcal{E}_{fathom} = \mathcal{O}\left( \sqrt{\frac{\sigma_L^2 + G^2}{m\overline{K}T}} + \sqrt[3]{\frac{\sigma_L^2}{\overline{K}T^2}} + \sqrt[3]{\frac{G^2}{T^2}} \right) \tag{16}$$

274 *with the following conditions:* $\overline{\eta}_L = \min\left(\sqrt{\frac{2\beta_0 mD}{\beta_1 \overline{K}LT(\sigma_L^2+G^2)}}, \sqrt[3]{\frac{\beta_0 D}{2.5\beta_2 \overline{K}^2 L^2 \sigma_L^2 T}}, \sqrt[3]{\frac{\beta_0 D}{2.5\beta_3 \overline{K}^3 L^2 G^2 T}}\right)$

275 *and $\eta_{L,t} \le 1/L$ for all $t$, where*

$$\overline{\eta}_L \triangleq \frac{1}{T}\sum_{t=1}^{T}\eta_{L,t} \qquad and \qquad \overline{K} \triangleq \frac{1}{T}\sum_{t=1}^{T}K_t \tag{17}$$

276 *and where*

$$\beta_0 = \frac{\sum_t \eta_{L,t}K_t}{T[\frac{1}{T}\sum_t \eta_{L,t}][\frac{1}{T}\sum_t K_t]} \ , \ \ \beta_1 = \frac{\sum_t \eta_{L,t}K_t\left[\frac{1}{T}\sum_t \eta_{L,t}\right]}{\sum_t \eta_{L,t}^2 K_t} \tag{18}$$

$$\beta_2 = \frac{\sum_t \eta_{L,t}K_t\left[\frac{1}{T}\sum_t \eta_{L,t}\right]^2\left[\frac{1}{T}\sum_t K_t\right]}{\sum_t \eta_{L,t}^3 K_t^2} \ , \ \ \beta_3 = \frac{\sum_t \eta_{L,t}K_t\left[\frac{1}{T}\sum_t \eta_{L,t}\right]^2\left[\frac{1}{T}\sum_t K_t\right]^2}{\sum_t \eta_{L,t}^3 K_t^3} \tag{19}$$

277 *We leave the proof in the Appendix beginning in eq(29).*

278 The values of $\beta_0$, $\beta_1$, $\beta_2$, $\beta_3$, and $\beta_4$ are dependent on the relative changes over the adaptive process
279 of these components, according to Chebyshev's Sum Inequalities (Hardy et al. [1988]). A special
280 case is when these quantities equal to 1 when both $\eta_{L,t}$ and $K_t$ are constant, which recovers the
281 standard upperbound for FedAvg from eq(16).

282 **Remark 2.** *The definitions in eq(17) combined with the conditions for $\overline{\eta}_L$ above is called the relaxed*
283 *conditions in this paper for the hyperparameters $\eta_{L,t}$ and $K_t$. The values of $\eta_{L,t}$ and $K_t$ are adaptive*
284 *during the optimization process between rounds $t = 1$ and $t = T$, as long as the above conditions are*
285 *satisfied for the guarantee in eq(31) to hold. This relaxation presents opportunities for a scheme such*
286 *as FATHOM to exploit for performance gain. For example, suppose $T$ approaches $\infty$ for a prolonged*
287 *training session. Then $\overline{\eta}_L$ would necessarily be sufficiently small for $\mathcal{E}_{fathom}$ to be bounded by*
288 *eq(16). However, for early rounds i.e. small $t$ values, $\eta_{L,t} \le T\overline{\eta}_L$ can be reasonably large and still*
289 *can satisfy eq(17), for the benefit of accelerated learning and convergence progress early on. Similar*
290 *strategy can be used for number of local steps to minimize local computations towards later rounds.*
291 *In any case, these strategies are mere guidelines meant to remain within the worst case guarantee.*
292 *However, Theorem 3 offers the flexibility otherwise not available. We will now show the empirical*
293 *performance gained by taking advantage of this flexibility.*

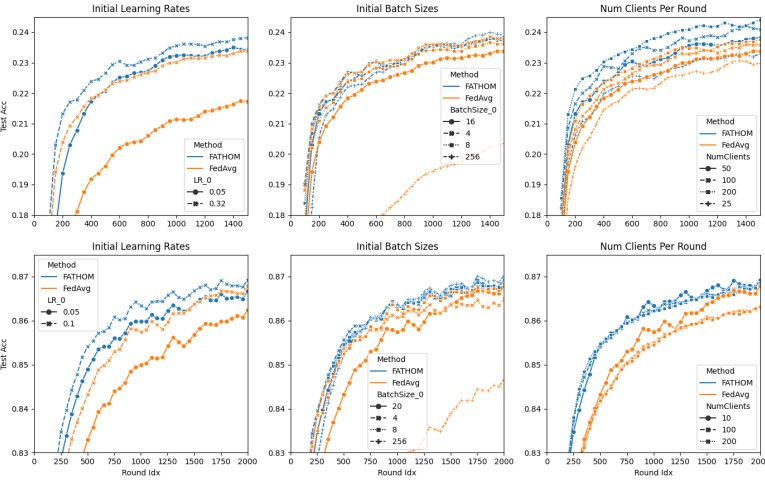

Figure 1: Test Accuracy Performance with various values of initial client learning rate (LR_0), initial batch size (BatchSize_0), and number of clients per round (NumClients). Top row: FSO sims. Bottom row: FEMNIST sims. Baseline values for FEMNIST: LR_0=0.1, BatchSize_0=20, NumClients=10. Baseline values for FSO: LR_0=0.32, BatchSize_0=16, NumClients=50.

## 5 Empirical Evaluation and Numerical Results

We present an empirical evaluation of FATHOM proposed in Section 3 and outlined in Algorithm 1. We conduct extensive simulations of federated learning in character recognition on the federated EMNIST-62 dataset (FEMNIST) (Cohen et al. [2017]) with a CNN, and in natural language next-word prediction on the federated Stack Overflow dataset (FSO) (TensorFlow-Federated-Authors [2019]) with a RNN. We defer most of the details of the experiment setup in Appendix Section C.1. Our choice of datasets, tasks and models, are exactly the same as the "EMNIST CR" task and the "SO NWP" task from Reddi et al. [2020]. See Figure 1 and Table 1 and their captions for details of the experiment results. Our evaluation lacks comparison with a few one-shot FL HPO methods discussed earlier in the paper because of a lack of standardized benchmark (until FedHPO- B Wang et al. [2022] was published concurrently as this work) to be fair and comprehensive.

The underlying principle behind these experiments is evaluating the robustness of FATHOM versus FedAvg under various initial settings, to mirror realistic usage scenarios where the optimal hyperparameter values are unknown. For FATHOM, we start with the same initial hyperparameter values as FedAvg. The test accuracy progress with respect to communication rounds is shown in Figure 1 from these experiments. We also pick test accuracy targets for the two tasks. For FEMNIST CR we use 86% and for FSO NWP we use 23%. Table 1 shows a table of resource utilization metrics with respect to reaching these targets in our experiments, highlighting the communication efficiency as well as reduction in local computation from FATHOM in comparison to FedAvg. To our knowledge, we are the first to show gain from an online HPO procedure over a well-tuned equivalent procedure with fixed hyperparameter values.

The federated learning simulation framework on which we build our algorithms for our experiments is FedJAX (Ro et al. [2021]) which is under the Apache License. The server that runs the experiments is equipped with Nvidia Tesla V100 SXM2 GPUs.

Table 1: Resource utilization in communication and local computation to reach specified test accuracy target for each task. All evalutions are run for ten trials. Bold numbers highlight better performance. NA means target was not reached within 1500 rounds for FSO NWP and 2000 rounds for FEMNIST CR, in any of our trials. LR_0 is initial client learning rate, BS_0 is initial batch size, and NCPR is number of clients per round. All experiments use baseline initial values except where indicated. For clarification, M is used in place for "million", and K for "thousand".
Baseline_fso : (LR_0 = 0.32, BS_0 = 16, NCPR = 50)
Baseline_femnist : (LR_0 = 0.10, BS_0 = 20, NCPR = 10)

| Tasks | Experiments | Number of Rounds To Reach Target Test Accuracy | | Local Gradients Calculated To Reach Target Test Accuracy | |
|---|---|---|---|---|---|
| | | FATHOM | FedAvg | FATHOM | FedAvg |
| FSO NWP Target@23% | Baseline_fso | **562 ± 12** | 971 ± 11 | **85M ± 1.2M** | 124M ± 1.3M |
| | LR_0 = 0.05 | **871 ± 7** | NA | **138M ± 3.2M** | NA |
| | BS_0 = 4 | 758 ± 43 | **580 ± 18** | 93M ± 2.8M | **74M ± 2.5M** |
| | BS_0 = 256 | **801 ± 28** | NA | **174M ± 18M** | NA |
| | NCPR = 25 | **970 ± 49** | 1283 ± 33 | **63M ± 2.7M** | 82M ± 3.8M |
| | NCPR = 200 | **396 ± 17** | 684 ± 26 | **280M ± 45M** | 350M ± 13M |
| FEMNIST CR Target@86% | Baseline_femnist | **739 ± 24** | 1098 ± 15 | **1.5M ± 36K** | 2.2M ± 64K |
| | LR_0 = 0.05 | **905 ± 21** | 1574 ±19 | **1.7M ± 28K** | 3.1M ± 28K |
| | BS_0 = 4 | **708 ± 17** | 885 ± 41 | **1.2M ± 28K** | 1.7M ± 88K |
| | BS_0 = 256 | **736 ± 20** | NA | **2.0M ± 44K** | NA |
| | NCPR = 100 | **777 ± 16** | 1436 ± 18 | **22M ± 0.27M** | 28M ± 0.39K |
| | NCPR = 200 | **790 ± 16** | 1481 ± 33 | **57M ± 1.0M** | 59M ± 1.3M |

## 6 Conclusion and Future Work

In this work, we propose FATHOM for adaptive hyperparameters in federated optimization, specifically for FedAvg. We analyze theoretically and evaluate empirically its potential benefits in convergence behavior as measured in test accuracy, and in reduction of local computations, by automatically adapting the three main hyperparameters of FedAvg: client learning rate, and number of local steps via epochs and batch size. An example of future efforts to extend this work is using a standardized benchmark such as Wang et al. [2022] for performance comparison against other FL HPO methods.

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
