# OpenReview forum: "Federated Hypergradient Descent"
_NeurIPS.cc/2022/Conference — NeurIPS 2022 Submitted_

### Official Review · Reviewer_KttA · 2022-07-09

**Rating:** 5
**Confidence:** 3
**Soundness:** 2 fair
**Presentation:** 3 good
**Contribution:** 2 fair

**Summary:**

This paper proposes a method for tuning the client step-size, number of epochs, and batch-size of FedAvg in a oneshot manner. They propose hypergradient-style updates to these quantities that can be done simultaneously with updating the parameters. The algorithm is evaluated on FEMNIST and StackOverflow.

**Questions:**

Notes:
1. 12: what does “open-boxed” mean?
2. 56: having difficulty following what this sentence is saying
3. 68: what is a “relaxed condition”?
4. 98: the two papers use different models and subsets of the data and so their settings are incomparable
5. 129: I suggest using \mathbb R instead of \mathcal R to denote the set of real numbers
6. 150: what do A, B, and I represent? also B was used earlier to denote the batch size
7. 169: I do not believe either paper states this as their objective
8. Theorem 1: this theorem also requires Assumption 1
9. Theorem 2: this theorem requires Assumptions 1 and 2
10. 225: the choice of using the bias is itself a hyperparameter
11. 263: this is an online convex optimization method? My understanding was the function f itself is non-convex.

**Limitations:**

The authors address the limitation of only being able to tune a few hyperparameters. Other limitations, especially the theoretical assumptions made and in the empirical evaluation, are less-discussed.

**Strengths And Weaknesses:**

**Post-rebuttal:**
Following clarification from the authors, I have decided to raise my score slightly but still cannot fully endorse this submission. Overall I think it is somewhat unsatisfactory to have to make such a strong assumption (convexity of the loss w.r.t. the batch-size in the sense of Murota (1990)) just to obtain subgradients. In particular, what is a bit jarring is the fact that it is still unclear whether there is *any* scenario under which this assumption holds. Standard regularity assumptions, while restrictive, at least come with specific model classes and loss functions under which they hold. Here are some things that would strengthen the theoretical component of the work:
1. Including a statement of the definition from Murota (1998) in the paper explicitly and showing where it is used in the proofs.
2. Defining subgradients for non-convex functions in the discrete setting, as they can be for the continuous setting (Clarke, 1990).
3. Providing an explicit setting where Assumption 2 holds.
4. Providing experimental evidence that Assumption 2 holds in practice. In particular, this type of convexity should be easy to measure because the underlying set is not uncountable.

Summary:
The paper is reasonably well-motivated, and the approach of targeting a few hyperparameters seems sensible. The experimental results indeed seem to demonstrate the utility of the proposed method. However, I have strong concerns regarding the assumptions the theoretical results and other concerns regarding the experimental comparison and discussion of past work that make me hesitant to accept the paper.

Strengths:
1. The paper tackles an important and timely problem in federated learning.
2. The authors devote a good deal of time to the communication and local computation complexity of their approach, both in the method development and empirical comparison sections.
3. The method is evaluated on relevant benchmarks in computer vision and NLP.
4. Code is included.

Weaknesses:
1. The derivation of the hypergradients lack a lot of detail. Theorem 1 introduces a great deal of machinery to try and define a sub-gradient w.r.t. a continuous surrogate variable of a discrete function. How the piecewise function L corresponds to the objective, what l represents precisely (rather than in words), and how Theorem 1 is used in the later proofs should be explained more precisely. The derivation of the gradients for the batch size and number of epochs in (14) and (15) is also not thoroughly explained.
2. Assumption 2, used by all three theorems in the paper, seems extremely strong. Is there at least an example of a function class in which this holds? Otherwise it seems the paper is avoiding the need to take derivatives through local iterations by optimizing an upper bound, and also hiding this in an assumption without discussing whether it is reasonable.
3. The experimental evaluation does not compare to baselines other than FedAvg, such as general hyperparameter tuning methods or the FL-specific methods discussed in the related work.
4. Some of the discussion of past federated hyperparameter optimization work is incorrect (see notes).
5. The method only works for tuning three hyperparameters. The authors do acknowledge this limitation, and it is indeed reasonable to focus on specific hyperparameters, e.g. if they are especially important. However, it would be good to have a discussion of whether the ones chosen here are indeed the most impactful.

&nbsp;
**References:**
1. Clarke. Optimization and nonsmooth analysis. SIAM, 1990.
2. Murota. Discrete convex analysis. Mathematical Programming, 1998.

---

> ### Author Response · Authors · 2022-08-02
> **Replies to Reviewer KttA**
>
> We would like to extend our sincere gratitude and appreciation to Reviewer KttA for their valuable insight, expertise, and most of all your time, for reviewing our paper and offering questions / comments.  We have considered all the comments carefully.  Below are our answers to each of the questions.  We have also submitted a revised paper / appendix, to reflect the changes cited below.
>
> > Weakness of Assumption 2 (convexity wrt hyperparameters)
>
> **[Authors' reply]** We have added Remark 1 (lines 118-121) to justify this assumption.
>
> >12: what does “open-boxed” mean?
>
> **[Authors' reply]** We have modified the wording from "open-boxed" to "we have full knowledge of the components / procedure" or something similar throughout the revised version which we have submitted.  We have initially used the term to mean that our algorithm does not use closed-boxed techniques such as many of the algorithms cited in the paper ([1], [2], etc), but since than have agreed that the term may create more questions than answers without full explanations.
>
> >56: having difficulty following what this sentence is saying
>
> **[Authors' reply]** Thanks for asking.  There was a typo within a confusing sentence in the Introduction section. Since then, we have removed a few confusing sentences from our revised version.
>
> >68: what is a “relaxed condition”?
>
> **[Authors' reply]**  We have added Remark 2  (lines 279 - 290), to dedicate to interpretations/insights regarding what we call the "relaxed condition", regarding how the additional components can potentially lead to better convergence than standard FedAvg.
>
> >98: the two papers use different models and subsets of the data and so their settings are incomparable
>
> **[Authors' reply]** Thanks for catching this mistake.  We had assumed they were standard benchmarks and thus were identical, but upon checking the code from [3] (https://github.com/mkhodak/FedEx/blob/main/femnist.py), it is indeed different from [4].  We have removed this portion of the paragraph in our revised version.
>
> >129: I suggest using \mathbb R instead of \mathcal R to denote the set of real numbers
>
> **[Authors' reply]** Done.
>
> >150: what do A, B, and I represent? also B was used earlier to denote the batch size
>
> **[Authors' reply]** Thank you for catching this error.  We have changed A to U, and B to W, between lines 132 and 133.
>
> >169: I do not believe either paper states this as their objective
>
> **[Authors' reply]** In [3], please see eq(4) (https://arxiv.org/pdf/2106.04502.pdf), which states exactly this objective.  In [5], please see Section 3.3 under subsection "Evaluating the loss of the aggregate model", where it says "The server maintains a small validation set on which it evaluates the loss."  In any case, this objective merely denotes the use of a held-out dataset to evaluate the model for training purpose, as an alternative to the use of the training set for the same goal of a one-shot procedure.
>
> >Theorem 1: this theorem also requires Assumption 1
>
> **[Authors' reply]** We believe that Assumption 1 is not needed for Theorem 1 per se, even though Assumption 1 is implicitly implied throughout our work. This theorem is about taking an additional fractional local step.  For example, we do not need to make the claim of $g_i(x)$ being unbiased estimate of $\nabla f(x)$.  Please let us know if this is not clear.
>
> >Theorem 2: this theorem requires Assumptions 1 and 2
>
> **[Authors' reply]** We believe we can make a similar argument of Theorem 2 not needing Assumption 1 per se, even though Assumption 1 is implicitly implied throughout our work.  Assumption 2 is needed which is stated in the paper.
>
> >225: the choice of using the bias is itself a hyperparameter
>
> **[Authors' reply]** We have agreed to remove this sentence from our revised submission.
>
> >263: this is an online convex optimization method? My understanding was the function f itself is non-convex.
>
> **[Authors' reply]** Thank you for raising this concern.  We have added Remark 1 (lines 118-121) to justify this assumption.  Basically it is needed to derive the subgradients in Theorems 1 and 2.  However, empirically the proposed algorithm works in the non-convex setting.
>
> ### **References**
> 1. Y. Zhou, P. Ram, T. Salonidis, N. Baracaldo, H. Samulowitz, and H. Ludwig. Single-shot hyper-parameter optimization for federated learning, 2022.
> 2. Z. Dai, K. H. Low, and P. Jaillet. Federated bayesian optimization, 2020.
> 3. M. Khodak, R. Tu, T. Li, L. Li, N. Balcan, V. Smith, and A. Talwalkar. Federated hyperparameter tuning: Challenges, baselines, and connections to weight-sharing. Beygelzimer, Y. Dauphin, P. Liang, and J. W. Vaughan, 2021.
> 4. S. Reddi, Z. Charles, M. Zaheer, Z. Garrett, K. Rush, J. Konecˇný, S. Kumar, and H. B. McMahan. Adaptive federated optimization, 2020.
> 5. H. Mostafa. Robust federated learning through representation matching and adaptive hyper-parameters, 2019.

---

> > ### Comment · Reviewer_KttA · 2022-08-06
> > **Further questions**
> >
> > Thank you for the reply. Some of my concerns remain, especially with the use of Assumption 2 as justification for the method. Here are a few comments about your responses.
> >
> > 1. Assumption 2 remains a strong and strange one to make. For one, how can a function be convex w.r.t. the batch size, which as an integer is not defined over a convex set (the assumption appears before any relaxation of K done later)? Second, if all you need it for is sub-gradients why not assume something like smoothness? Is there empirical evidence that can be provided to justify this?
> > 2. While the objective is stated in [3] the one they actually used is in their Equation 7. For [5], maintaining a separate validation set does not imply the same objective of adding the losses. Regardless of the point being made it is important to be precise.
> > 3. Since you are using different assumptions in different places it is useful to state each assumption you are making in the theorem statement.

---

> > > ### Author Response · Authors · 2022-08-08
> > > **Reply to your remaining concerns**
> > >
> > > Thank you again for your suggestions and for raising your concerns, which have been tremendously helpful in improving our paper.  We have submitted a newly revised version since your latest feedback.  The following are our answers to your concerns.  Again thank you for your time.
> > >
> > > > Assumption 2 remains a strong and strange one to make. For one, how can a function be convex w.r.t. the batch size, which as an integer is not defined over a convex set (the assumption appears before any relaxation of K done later)? Second, if all you need it for is sub-gradients why not assume something like smoothness? Is there empirical evidence that can be provided to justify this?
> > >
> > > **[Authors' reply]** Thank you so much for this concern.  We have slightly modified Assumption 2 in the latest revision, by citing "Discrete Convex Analysis" by Murota et al, 1998 to formalize the definition of convexity defined in discrete domain, i.e. for functions $f : \mathcal{Z}^{n} \rightarrow \mathbb{R}$.  We have also added an illustration (figure 2 in Appendix A, fathom-supp.pdf) as an addendum to the proof of Theorem 1, to clarify our concept of convexity for K and how we use it to formulate our subgradients.  We do not assume smoothness for $K$, in fact, subgradient methods remove the requirements that $f$ be differentiable.  We hope these modifications to the paper clarify Assumption 2.  We concede that we do not have empirical evidence for Assumption 2 per se, even though we rely on the empirical results from our experiments to validate our algorithm.  Our reasoning draws analogy from the success of gradient descent, that its convergence requirements such as smoothness is only useful for a guarantee but which belie its empirical success, where most of the time we would have no idea what the L factor is for our neural network that we run gradient descent methods on.  In our case, it is still important to be able to prove convergence of our algorithm by making reasonable assumptions, even though we know they are not always practical in our experiments, as long as we understand the limitations of these assumptions.  And your feedbacks have helped us tremendously on this.
> > >
> > > > While the objective is stated in [3] the one they actually used is in their Equation 7. For [5], maintaining a separate validation set does not imply the same objective of adding the losses. Regardless of the point being made it is important to be precise.
> > >
> > > **[Authors' reply]** We have agreed that our claims about [3] and [5] may have been misleading and these sentences do not benefit the paper in any way.  We have removed these sentences in the newest revision.
> > >
> > > > Since you are using different assumptions in different places it is useful to state each assumption you are making in the theorem statement.
> > >
> > > **[Authors' reply]** We have modified and added the phrase "We claim, under Assumption 2," to Theorems 1 and 2, lines 179 and 199, to add clarity to the claims of these theorems.  Thank you for the great suggestion.

---

> > > > ### Comment · Reviewer_KttA · 2022-08-09
> > > > **Response**
> > > >
> > > > Okay, thank you for the update; please see the comments in my updated review.

---

> > > > > ### Author Response · Authors · 2022-08-09
> > > > > **Thank you so much**
> > > > >
> > > > > Thank you so much for your time and your insightful feedback!

---

### Official Review · Reviewer_bwW1 · 2022-07-11

**Rating:** 5
**Confidence:** 1
**Soundness:** 3 good
**Presentation:** 3 good
**Contribution:** 3 good

**Summary:**

I am not an expert in this area and do not know anything about federated learning. The last time I worked directly with hyperparameter optimization approaches was in 2017 but was focused on offline hyperparameter optimization and not in the context of federated learning. As such, my review is not very insightful.

---

The paper proposes an approach for online hyperparameter optimization in a federated learning setting. The proposed approach (FATHOM) is more communication efficient than previous approaches and achieves better results than previous approaches with manually fine-tuned hyperparameters. However, since it assumes access to gradients it is restricted to a certain set of hyperparameters that can be optimized and the paper evaluates the approach by optimizing the client learning rate, number of local steps, and batch size. Compared to a baseline (FedAvg) the proposed approach is more efficient and obtains better final test accuracy.

**Questions:**

I can't comment on the evaluation protocol itself since I don't know the related work. The target accuracy numbers seem somewhat arbitrary (86% and 23%) but may be standard for this kind of evaluation.

**Limitations:**

The paper itself mentions several limitations, such as applicability only to specific hyperparameters for which gradients can be obtained.

**Strengths And Weaknesses:**

The paper motivates its approach well and clearly states its advantages and limitations compared to other approaches. The comparison to FedAvg is based on different hyperparameter initializations and shows improved results on most of them.
I did not check the theoretical convergence part.
The evaluation of the approach seems clear.

---

> ### Author Response · Authors · 2022-08-02
> **Replies to Reviewer bwW1**
>
> We would still like to extend our thanks to Reviewer bwW1 for their minimalist review.  We have since submitted a revised paper based on feedbacks from other reviewers.  If you would be so kind, we invite you to take a second look at the improved version.

---

### Official Review · Reviewer_K7kJ · 2022-07-12

**Rating:** 6
**Confidence:** 3
**Soundness:** 3 good
**Presentation:** 3 good
**Contribution:** 3 good

**Summary:**

This paper proposes to optimize the hyperparameters and parameters of a federated learning model jointly via gradient descent in an online fashion. To achieve this, the paper extends previous works on hypergradient to derives analytical gradients with respect to the local learning rate and the number of gradient steps. The paper analyzes the convergence of the proposed method and verifies the empirical improvement over FedAvg using real-world FL experiments.

**Questions:**

I've listed a few questions and comments as "Weaknesses" in the section above.

**Limitations:**

A limitation and potential negative societal impact are both discussed.

**Strengths And Weaknesses:**

Strengths:
- The proposed method is able to jointly optimize the hyperparameters and weight parameters of an FL model in a one-shot online manner, which is a natural and promising direction.
- The approach taken by the paper is reasonable: derive the gradient w.r.t. the hyperparameters of interest, and then use gradient descent to optimize them together with model parameters.

Weaknesses:
- A few related works are missing which I have listed below. Reference [*a*] aims at single-shot hyperparameter optimization for FL, i.e., they aim to identify a single set of hyperparameters which work well for multiple applications, and hence should be cited in the first paragraph of Section 2; [*b*] also used Bayesian optimization for hyperparameter optimization in FL and hence should be added to lines 114-116; [*c*] provided a benchmark for federated hyperparameter optimization.
[*a*] Single-shot Hyper-parameter Optimization for Federated Learning: A General Algorithm & Analysis, 2022.
[*b*] Differentially Private Federated Bayesian Optimization with Distributed Exploration, NeurIPS 2021.
[*c*] FedHPO-B: A Benchmark Suite for Federated Hyperparameter Optimization, 2022.
- In the experiments, the proposed method is only compared with vanilla FedAvg, which may not be enough especially since a number of related works on single-shot federated hyperparameter optimization have been listed in Section 2. Even if comparisons are not possible, some justifications should be given.
- Theorem 3: instead of listing the detailed expressions of the entire theorem, I would prefer to see some interpretations/insights regarding the theoretical results. For example, how do the additional components introduced by the proposed algorithm (i.e., adaptive tuning of $\eta$ and $K$) affect the convergence of FedAvg? Are there conditions under which these additional components can lead to better convergence then standard FedAvg according to Theorem 3?
- Perhaps partially due to my unfamiliarity with gradient-based optimization, I feel that some of the technical details should be given more explanations. Firstly, Assumption 2 may need some further explanations and justifications. For example, $x_t$ and $\mathbb{E}_t$ haven't been introduced till this point. A justification should be given as to how realistic this assumption is. Secondly, in Equation (5), some more explanations on how the last $\approx$ is derived are needed. Next, in equations (14) and (15), why are these two equations designed in this way?
- The paper has many typos, for example:
-- line 34: a -> an
-- line 56: justify -> justifying; a -> an
-- line 104: directing -> directly
-- line 145: $K$ should be $K_i$?

---

> ### Author Response · Authors · 2022-08-02
> **Replies to Reviewer K7kJ**
>
> We would like to extend our sincere gratitude and appreciation to Reviewer K7kJ for their valuable insight, expertise, and most of all your time, for reviewing our paper and offering questions / comments.  We have considered all the comments carefully.  Below are our answers to each of the questions.  We have also submitted a revised paper / appendix, to reflect the changes cited below.
>
>
> >* A few related works are missing which I have listed below...
>
> **[Authors' reply]** We have included these works in the newest revision.  We are particularly excited about the new benchmark suite, FedHPO-B.  Thank you.
>
> >* In the experiments, the proposed method is only compared with vanilla FedAvg, which may not be enough especially since a number of related works on single-shot federated hyperparameter optimization have been listed in Section 2. Even if comparisons are not possible, some justifications should be given.
>
> **[Authors' reply]** Justifications have been given in Section 5 "Empirical Evaluations and Numerical Results", lines 301 - 303: Our evaluation lacks comparison with a few one-shot FL HPO methods discussed298
> earlier in the paper because of a lack of standardized benchmark (until FedHPO- B Wang et al. [2022])
> was published concurrently as this work) to be fair and comprehensive.
>
> >* Theorem 3: instead of listing the detailed expressions of the entire theorem, I would prefer to see some interpretations/insights regarding the theoretical results. For example, how do the additional components introduced by the proposed algorithm (i.e., adaptive tuning of  and ) affect the convergence of FedAvg? Are there conditions under which these additional components can lead to better convergence then standard FedAvg according to Theorem 3?
>
> **[Authors' reply]** We have added Remark 2 (lines 281 - 292), to dedicate to interpretations/insights regarding the theoretical results, regarding how the additional components can potentially lead to better convergence than standard FedAvg.  Thank you for your suggestion.
>
> >* Perhaps partially due to my unfamiliarity with gradient-based optimization, I feel that some of the technical details should be given more explanations. Firstly, Assumption 2 may need some further explanations and justifications. For example,  and  haven't been introduced till this point. A justification should be given as to how realistic this assumption is. Secondly, in Equation (5), some more explanations on how the last  is derived are needed. Next, in equations (14) and (15), why are these two equations designed in this way?
>
> **[Authors' reply]** You are absolutely correct about Assumption 2.  We have slightly modified the wording of Assumption 2 and added Remark 1 (lines 158 - 161) to justify this assumption: "Assumption 2 is necessary to guarantee the existence of subgradients derived in Theorems 1 and 2, and it will be assumed for this work. In problems dealing with deep neural networks, it is reasonable to not assume convexity w.r.t. hyperparameters. However, from our empirical results, we claim that the proposed algorithm is still able to operate as desired under this condition."  You are also spot on regarding equations (5), (14), and (15).  We have since added explanations to facilitate the clarity and readability of the paper.  Please see lines 168 - 170 for eq(5), lines 238 - 241 for eqs(14) and (15), in the latest revised paper.  Thank you.
>
> >* The paper has many typos, for example:
> -- line 34: a -> an -- line 56: justify -> justifying; a -> an -- line 104: directing -> directly -- line 145:  should be ?
>
> **[Authors' reply]** Thank you so much for these.  You are absolutely correct on the rules regarding indefinite articles for abbreviations and acronyms, as laid out here https://www.languageediting.com/how-to-choose-the-indefinite-article-a-or-an-before-acronyms/.  We have since corrected these mistakes.

---

> > ### Comment · Reviewer_K7kJ · 2022-08-09
> > **Thank you for the reply**
> >
> > Thank you for the reply. I've read the authors' response and the other reviewers' review, and I'd like to keep my current rating.

---

> > > ### Author Response · Authors · 2022-08-09
> > > **Thank you again**
> > >
> > > Thank you again for your time.

---

### Meta-Review · Area_Chair_Msfv · 2022-08-22

**Recommendation:** Reject
**Confidence:** Certain

**Metareview:**

We thank the authors and the reviewers for their involvement in this interactive reviewing process. While the paper clearly generated interest and brings new notions to the community, we felt that a major revision is necessary before the paper can be a successfull NeurIPS submission. Consequently, we unfortunately recommend rejection. Points that stood out are

1. at least one essential assumption is simultaneously strong, hard to check in practice, and not appropriately discussed in the original version;
2. unusual but central notions like discrete convexity and hypergradients are not appropriately introduced. This made important technical parts of the paper hard to proofread;
3. lack of some natural baselines.

We hope that the detailed discussions will be useful in further improving the manuscript.




**Award:**

No

---

### Decision · Program_Chairs · 2022-09-14

Reject